# Construction of a Collagen-like Protein Based on Elastin-like Polypeptide Fusion and Evaluation of Its Performance in Promoting Wound Healing

**DOI:** 10.3390/molecules28196773

**Published:** 2023-09-23

**Authors:** Yingli Chen, Yuanyuan Wu, Fengmin Xiong, Wei Yu, Tingting Wang, Jingjing Xiong, Luping Zhou, Fei Hu, Xianlong Ye, Xinmiao Liang

**Affiliations:** 1Ganjiang Chinese Medicine Innovation Center, Nanchang 330100, China; chenyingli@jcmsc.cn (Y.C.); wuyuanyaun@jcmsc.cn (Y.W.); xiongfengmin@jcmsc.cn (F.X.); yuwei@jcmsc.cn (W.Y.); wangtingting@jcmsc.cn (T.W.); xiongjingjing@jcmsc.cn (J.X.); zhouluping@jcmsc.cn (L.Z.); hufei@jcmsc.cn (F.H.); 2Key Laboratory of Separation Science for Analytical Chemistry, Dalian Institute of Chemical Physics, Chinese Academy of Sciences, Zhongshan Road 457, Dalian 116023, China

**Keywords:** human-like collagen (hCol), elastin-like polypeptide (ELP), fusion protein, wound healing

## Abstract

In the healing of wounds, human-like collagen (hCol) is essential. However, collagen-based composite dressings have poor stability in vivo, which severely limits their current therapeutic potential. Based on the above, we have developed a recombinant fusion protein named hCol-ELP, which consists of hCol and an elastin-like peptide (ELP). Then, we examined the physicochemical and biological properties of hCol-ELP. The results indicated that the stability of the hCol-ELP fusion protein exhibited a more compact and homogeneous lamellar microstructure along with collagen properties, it was found to be significantly superior to the stability of free hCol. The compound hCol-ELP demonstrated a remarkable capacity to induce the proliferation and migration of mouse embryo fibroblast cells (NIH/3T3), as well as enhance collagen synthesis in human skin fibroblasts (HSF) when tested in vitro. In vivo, hCol-ELP demonstrated significant enhancements in healing rate and a reduction in the time required for scab removal, thereby exhibiting a scar-free healing effect. The findings provide a crucial theoretical foundation for the implementation of an hCol-ELP protein dressing in fields associated with the healing of traumatic injuries.

## 1. Introduction

The skin, which is made up of the epidermis, dermis, and subcutaneous tissue, is the body’s largest organ system [1]. The integrity of healthy skin has three main functions: a barrier function, a sensory function, and a metabolic function [2]. Physiological regulation of skin wound healing is a complex and overlapping process; it is mainly composed of four phases: the hemostasis, inflammatory, proliferative, and remodeling phases [3]. However, there are many factors that affect wound healing, such as age, medication, obesity, alcohol consumption, smoking, nutrition, etc., [4]. The most advanced treatments for wound healing are gels rich in the patient’s platelet plasma (PRP) [5,6]; however, this treatment is expensive and has some side effects. According to current reports, collagen has the potential to accelerate wound healing and elastin has the effect of preventing scar proliferation [7,8]. Skin transplantation rich in collagen and elastin can treat extensive skin wounds [9].

The majority of the protein in the human body, between 25% and 30% of the total, is made up of collagen. It is the primary element of the extracellular matrix (ECM) and is crucial for maintaining the physiological functions of cells, tissues and organs, and damage repair [10,11,12]. There are currently 29 different forms of collagen that have been identified, and they are all composed of the repeated Gly-X-Y domains that allow collagen to form a triple helix structure [13,14]. Collagen has a unique molecular structure that gives it a strong affinity for certain tissues as well as excellent biological activity, biocompatibility, and biodegradability. It is widely employed in many industries, including biomedicine, cosmetics, food, and others [15,16,17].

As a result of advancements in science and technology, researchers have extensively studied natural materials, including those containing collagen, in the field of skin tissue engineering. These materials have been found to promote angiogenesis and facilitate full-layer wound healing [18]. They also encourage the organized alignment of collagen fibers in newly formed dermal tissue and possess the ability to repair skin wounds and expedite the healing process [19]. Lin H and Xu L et al. reported the positive effects of collagen peptides or collagen-composite materials in inhibiting the inflammatory response, promoting fibroblast infiltration, new blood vessel formation, and collagen deposition when used in animal models of trauma [19,20]. These findings confirm the essential need for collagen in skin regeneration and wound healing. Moreover, dressings containing collagen have been widely used in clinical practice, and they aid in the regeneration of dermal tissue in skin deformities and wounds. Autoskin grafts frequently employ products such as Integra^®^ (LifeSciences Corporation, Princeton, NJ, USA) and MatriDerm^®^ (Dr. Otto Suwelack Skin & Health Care AG, Billerbeck, Germany). These products are primarily made of animal collagen. They promote wound bed regeneration and wound healing by improving collagen, pigmentation, and vascularization [21,22,23]. Moreover, collagen can significantly reduce scar contracture and the risk of dysfunction. However, the main component of the collagen products commonly used in clinical practice is animal-derived collagen [24,25], and the US Food and Drug Administration clearly states that given the narrow safety pathways that are implemented in manufacturing, animal-derived materials maybe pose a risk of transmitting viruses and spongiform encephalopathy [26]. For example, the previous outbreak of the mosquito-borne flavivirus Zika virus (ZIKV) has posed a huge threat to the safety of biological agents [27]. Dressings that solely consist of collagen exhibit subpar mechanical properties and a faster degradation rate, thereby significantly impacting the practicality of collagen utilization [28,29]. Consequently, there exists an urgent requirement to develop efficient and reliable collagen excipients, and numerous endeavors are currently underway to address this issue.

The primary purpose of elastin, an intrinsically disordered protein found in the extracellular matrix, is to provide biological connective tissues with mechanical qualities such as elasticity and durability [30,31]. Elastin-like polypeptides (ELPs) are a derivative of pro-elastin. Their unique properties make them attractive materials for a variety of biomedical applications [32,33]. Studies have shown that ELPs can be expressed as a fusion molecule with various bioactive proteins or peptides, resulting in chimeric fusion proteins that retain the function of each component, where they can use the critical solution temperature phase behavior to simply and quickly purify proteins [34,35]. In addition, ELP-based fusion proteins have been shown to protect biomolecules from proteolysis [36] and can act as a “drug depots” to extend the half-life of drugs in the body [37]. Importantly, owing to stimulus-responsive self-assembly and the ideal mechanical properties of ELPs, they can be used to develop a variety of different matrices for wound treatment. Yeboah and Devalliere et al. showed that the fusion of growth factors (e.g., SDF1 and KGF) with ELPs increased their stability in the chronic wound and promoted wound healing to a great extent compared with non-fusion proteins. Therefore, ELP fusion proteins have great potential in wound healing [38,39].

In the present study, our hypothesis was that the utilization of hCol in combination with an ELP-based approach would yield superior results in the enhancement of wound healing. In consideration of this perspective, we developed a fusion protein molecule called hCol-ELP, with the aim of enhancing the stability and biological activity of hCol. Subsequently, we conducted an assessment of the overall physicochemical and biological properties of hCol-ELP. The findings suggest that hCol-ELP possesses the capability to expedite wound healing and suppress scar formation. The aforementioned statement establishes the theoretical basis for the potential clinical utilization of hCol-ELP in the context of skin wound healing.

## 2. Results

### 2.1. Construction and Expression of the hCol-ELP Fusion Protein

To enhance the synergistic effects of hCol and ELP in the process of wound healing, a fusion protein, hCol-(G_4_S)_3_-ELP (hCol-ELP), was created by linking ELP to the C-terminus of hCol using a (G_4_S)_3_ linker (Figure 1A). This linker, which is of considerable length, serves the purpose of preserving the spatial separation between the ELP and hCol functional domains. This spatial separation is crucial for ensuring the correct folding and maximal biological activity of the fusion protein [40]. In addition, we also generated hCol and ELP proteins as experimental controls. The SDS-PAGE analysis demonstrated that soluble forms of hCol, hCol-ELP, and ELP were successfully expressed in *E. coli* (Figure 1B). Therein, the molecular weights of hCol, hCol-ELP, and ELP were approximately 32 kDa, 35 kDa and 30 kDa, respectively, which were consistent with the theoretical molecular weights. The three proteins were produced with purities of 93.1%, 97.5%, and 99.1%, respectively, after undergoing salting-out treatment and one-step ion-exchange chromatography to purify the products, according to SDS-PAGE and SEC-HPLC analyses (Figure 1C). All the products, hCol, hCol-ELP and ELP, had a high purity level.

### 2.2. Characterization of the hCol-ELP Fusion Protein

The ultimate morphologies and structures of hCol, hCol-ELP, and ELP were characterized with microscopic techniques. As shown in Figure 2A, the surface morphologies of hCol, hCol-ELP, and ELP freeze-dried powders were almost the same, a white sponge shape, whereas each had different internal structures. The pure collagen hCol had a lamellar structure with a larger mesh, whereas the fusion protein hCol-ELP had a more compact and uniform lamellar microstructure. Next, the collagen properties of hCol and hCol-ELP were tested through collagenase digestion tests. Figure 2B demonstrates that the hCol and hCol-ELP samples were rapidly and completely digested within 0.5 h, whereas the ELP without collagen properties remained stable when incubated in a collagenase solution for 6 h at 37 °C. This indicates that hCol and hCol-ELP exhibited collagen characteristics, whereas ELP did not. In addition, we conducted a 4-day accelerated stability assessment in vitro, and SDS-PAGE was utilized to evaluate the extent of protein degradation. As shown in Figure 2C,D, compared with the stability of hCol, hCol-ELP and ELP showed stronger stability after incubation at 37 °C for 96 h. The increased stability of hCol-ELP suggests that fusion with ELP increases the stability of hCol. This was further confirmed by stability tests conducted in tissue fluid systems. We noted that the hCol-ELP fusion protein remained slightly degraded throughout the incubation period, whereas the target bands degraded more in hCol samples (Figure 2E,F). According to the above results, an hCol-ELP fusion protein that had collagen properties and was highly stable was successfully prepared.

### 2.3. The hCol-ELP Fusion Protein Possesses an Outstanding Capacity to Induce Cell Proliferation and Migration, as Well as to Promote Collagen Regeneration

To assess the activities of the proteins, we evaluated their effects on proliferation and wound closure in NIH/3T3 cells using a cell counting kit (CCK8) and assessed their pro-migration activity. The CCK8 assay results (Figure 3B) indicated that treatment with hCol-ELP resulted in excellent cell viability and a significant increase in the cell proliferation rate at both 4 and 8 μM concentrations compared with the control. The cells treated with hCol and ELP showed no apparent cell proliferation activity. The in vitro scratch assay results showed that the hCol, hCol-ELP, and ELP proteins significantly promoted wound closure through cell migration in NIH/3T3 cells (Figure 3A). Among them, hCol-ELP had the fastest fibroblast migration rate of 40.1 ± 5.84% in 24 h when compared with the hCol (36.9 ± 4.74%) and ELP (35.4 ± 3.40%) groups. The above results demonstrate that hCol-ELP possesses an outstanding capacity to induce the proliferation and migration of NIH/3T3 cells. The hCol and hCol-ELP proteins effects on type I collagen synthesis (*n* = 3) were evaluated using ELISA and RT-PCR assays (Figure 3C,D). The results show that hCol and hCol-ELP promoted the expression of pro-collagen I α1 in HSFs. However, the expression of pro-collagen I α1 was significantly higher in the hCol-ELP group than the hCol group. In addition, the RT-PCR results also suggested that hCol and hCol-ELP significantly increased the expression level of collagen I α1 mRNA and exhibited a similar trend to the expression of pro-collagen I α1. The results reveal that hCol-ELP has a superior effect than hCol in increasing the expression of collagen I.

### 2.4. The hCol-ELP Fusion Protein Accelerates Wound Healing In Vivo

To assess the bioactivity of the hCol-ELP fusion protein in wound healing, a sample of hCol-ELP was applied to the wound and secured with a 3M clear compress; hCol and ELP were used as the control groups. The closure of the wound was monitored and recorded on days 0, 3, 5, 7, 11, and 14 using a camera. As shown in Figure 4B,C, the hCol-ELP group had a significantly faster wound closure rate than the other groups 3 days after modeling *(p* _hCol-ELP_ vs. _model_ < 0.0001, *p*
_hCol-ELP_ vs. _hCol_ = 0.0407, *p* _hCol-ELP_ vs. _ELP_ = 0.0332). At 5 days post-wounding, the wound closure rate of the hCol-ELP group was still greater than in the model group (*p* _hCol-ELP_ vs. _model_ < 0.0001), but there was no significant difference between the hCol and ELP groups. In addition, we observed that the wounds treated with hCol-ELP closed faster than those in any other group during the observation period (Table 1). All of these results suggest that hCol-ELP can induce a higher wound healing efficiency in the early stages of post-wounding and maintain high bioactivity for an extended period of time.

### 2.5. The hCol-ELP Fusion Protein Improves the Healing Quality of the Wound Epidermis and Collagen Deposition

The effect of hCol-ELP on wound healing was further examined through histological analysis. As shown in Figure 5A,B, during the process of wound contraction and re-epithelialization in the early stages of wound healing, the hCol-ELP, hCol, and ELP groups exhibited partial healing with improved but disconnected granulation formation. In contrast, the model group displayed unhealed wounds with minimal granulation tissue formation. The widths of wound surfaces in the hCol-ELP group, hCol group, and ELP groups were significantly smaller than those in the model group (Table 2, Figure 5C), and the thicknesses of epidermises in the hCol-ELP, hCol, and ELP groups were significantly smaller than those in the model group (Table 2, Figure 5D). At 14 days post-wounding, the same trend among the groups remained. Furthermore, collagen formation was of great significance in wound healing. Masson staining showed that the skin of the mice in the hCol, hCol-ELP, and ELP groups had formed a dense collagen fiber layer on the 7th and 14th days after the skin wounding (Figure 5B–E). Compared with the other two groups, the collagen bundle deposition formed by the skin in the hCol-ELP group was more uniformly thick. At the same time, by observing the immunofluorescence staining of type I collagen in the skin tissues of mice on the 7th and 14th days after the defect (Figure 6A–C), we observed that type I collagen was more widely distributed in the skin tissues of mice in the hCol-ELP group than in the model, hCol, and ELP groups. All of these results confirm that the designed hCol-ELP fusion protein can promote the regeneration of wound tissue and the deposition of collagen.

### 2.6. Detection of Expression of Wound-Tissue-Related Factors via RT-PCR

The mRNA expression of several factors involved in wound healing was examined using RT-PCR. The results show that on the 7th and 14th days post-wounding, the mRNA levels of tumor necrosis factor-alpha (TNF-α), interleukin 6 (IL-6), and α smooth muscle actin (α-SMA) in the hCol, ELP, and hCol-ELP groups were downregulated compared with the model group (Figure 7A,B,D). In contrast, the mRNA level of collagen I alpha1 (Col1 α1) was upregulated (Figure 7C), especially in the hCol-ELP group, where there were statistically significant differences in the levels of all factors (*p* < 0.05); however, there were no statistically significant differences between the hCol, ELP, and hCol-ELP groups (*p* > 0.05).

These outcomes demonstrated the effectiveness of the recombinant fusion protein (hCol-ELP) that we created in mouse skin wound models. The protein was found to promote collagen regeneration, reduce inflammation, and expedite wound healing (Figure 8).

## 3. Discussion

Research shows that collagen can promote full-layer wound healing and angiogenesis. It also promotes the orderly arrangement of collagen fibers in new dermal tissue, repairs skin wounds, and improves wound healing quality [18,19]. Collagen is one of the most frequently used materials for skin wound healing [41]. However, dressings containing only collagen as the sole component have inferior mechanical properties and faster degradation rates [28,29]. Therefore, we designed and synthesized an hCol-ELP fusion molecule to enhance the mechanical properties, stability, and bioactivity of hCol. The degradation curves of hCol-ELP in PBS and tissue fluid systems confirmed that the stability of hCol-ELP was significantly better than that of free hCol (Figure 2). In addition, hCol-ELP possessed an outstanding capacity to induce NIH/3T3 and HSF cell proliferation and migration and promote collagen regeneration in vitro (Figure 3). Importantly, in the repair of mouse skin wounds, by achieving the functions of hCol and ELP, the hCol-ELP fusion molecule had a significant advantage in accelerating wound healing and avoiding the development of abnormal scars compared with the application of hCol or ELP alone (Figure 4).

In this study, we addressed the limitation of using pure collagen by using an ELP fusion protein. There have been relevant reports on ELP–collagen fusion, and compared to pure collagen scaffolds, ELP–collagen composites have been shown to exhibit superior mechanical and physical properties [29,42]. Therefore, by designing an hCol-ELP fusion protein, not only can the mechanical properties and stability of pure hCol be enhanced but ELP and hCol can act together to promote wound healing, which can help enhance the overall wound healing effect. On the basis of previous studies, we fused ELP to the C-terminal of hCol through a (G_4_S)_3_ linker to prepare the fusion protein hCol-(G_4_S)_3_-ELP (hCol-ELP) [43]. We observed that hCol-ELP had a lower rate of degradation than hCol (Figure 2E,F). This result verified that the hCol-ELP fusion protein had a high level of stability. However, there is still room for optimization in the design of the hCol-ELP fusion molecules. This includes the fusion method of ELP at the N or C terminus of the target protein, the amino acid composition and length of ELP, and the selection and length of the linker. These aspects can be further enhanced in subsequent experiments.

Nutrition is a fundamental requirement for wound healing, and creating a favorable nutritional environment is one of the most effective strategies to facilitate rapid wound healing [19]. The hCol-ELP fusion protein is a collagen–elastin-like fusion protein containing collagen; collagen plays a vital role as key component of the extracellular matrix and is essential for regulating wound healing, whether in its natural fibrillar conformation or as a soluble component in the wound environment [44]. Elastin-like protein helps restore a complete and functional elastic fiber network, thereby restoring complete skin function after injury [7]. Both collagen and elastin-like protein are major components of skin regeneration and repair and provide an essential nutrient matrix for wound healing. The hCol-ELP has a variety of biological activities. First, new collagen and/or elastic fibers can be synthesized after a false signal is sent out to the skin fibroblasts. Secondly, both collagen and fused ELP derivatives can promote keratinocyte migration and contribute to wound re-epithelialization [45,46,47]. In addition, collagen degradation can also contribute to wound inflammation, angiogenesis, and re-epithelialization [48]. In the course of the inflammatory phase, soluble fragments of collagen degradation attract immune cells, such as macrophages, which patrol the site to eliminate pathogens and deactivate damaged tissue. This facilitates the transition to the proliferative stage. In this study, the hCol-ELP fusion protein can be rapidly degraded into small molecular fragments by collagenase. It has been proven that full-length collagen can release bioactive fragments (also known as matrix inhibins), such as endostatin and tumstatin, through the action of collagenase. These fragments can specifically guide vascular pruning and thus rebuild tissue structure during the healing process [44]. Therefore, the advantage of using hCol-ELP to promote wound healing may be due to its ability to rapidly release active fragments upon decomposition, thereby enhancing tissue reconstruction, which is consistent with our observations in a mouse trauma model (Figure 4). Furthermore, according to previous studies, the fusion of ELPs can increase the cellular activity and promote the wound healing ability of its fusion molecules [38,48]. These results are consistent with the observations made when hCol-ELP was applied to NIH/3T3 and HSF cells. As shown in Figure 3, the hCol-ELP group had more significant effects on promoting cell migration, proliferation, and collagen regeneration than the hCol or ELP groups.

TNF-α and IL-6 are crucial inflammatory cytokines and markers of an inflammatory response. In this study, the levels of TNF-α and IL-6 on the wound surface were significantly reduced in the hCol, hCol-ELP, and ELP groups on the 7th and 14th days after injury (Figure 7A,B). It was worth noting that compared with the hCol and ELP groups, the hCol-ELP group had a more significant effect on day 7, indicating that hCol-ELP may have stronger anti-inflammatory activity in the early stages of trauma. The wound healing rate and HE staining sections confirmed this perspective (Figure 4 and Figure 5A,C). On the 3rd and 5th days after trauma, the wound healing rate of the hCol-ELP group was significantly higher than that of the other groups, and the width of the wound on the 7th day was also significantly smaller than in the other groups. This may be due to the significant anti-inflammatory function of hCol-ELP, which is beneficial for promoting rapid wound healing. In addition, 14 days after the injury, the epidermal level of the hCol-ELP group was lower than that of the model group, suggesting that hCol-ELP may have a better effect on inhibiting scar hyperplasia (Figure 5A,D). Recombinant human collagen type III has been widely used in the field of medicine and aesthetics and has a good therapeutic effect by promoting wound healing [49]; however, there are no specific reports on inhibiting scars. Currently, there are products that use elastin peptides to delay skin aging [50]. In addition, elastin plays an important role in preventing scar hyperplasia [7,8], we combined the functions of collagen and elastin to create an hCol-ELP fusion protein, the results show that the hCol-ELP fusion molecule has a good effect in terms of promoting wound healing and inhibiting scars.

The primary actions during the proliferation stage (approximately 3–10 days after the injury) include covering the wound surface, generating granulation tissue, and reestablishing the vascular network. As a result, local fibroblasts migrate along the fibrin network and re-epithelialization begins at the wound edge [51]. As depicted in Figure 5, during the initial phase of trauma, the hCol-ELP, hCol, and ELP groups exhibited partial wound healing with limited connective granulation formation. In contrast, the model group displayed unhealed wounds with minimal granulation tissue formation. At 14 days after the injury, significant wound healing and granulation formation were observed in all groups. Moreover, the formation of collagen is of great significance for wound healing. The histological analysis and q-PCR results (Figure 6 and Figure 7) showed that dense collagen fiber layers could be formed in the skin of the hCol, hCol-ELP, and ELP groups on the 7th and 14th days after the skin defect. Additionally, the collagen bundles formed in the hCol-ELP group were more uniform and thicker than those in the other two groups. These results confirmed that the designed hCol-ELP fusion protein can reduce wound tissue regeneration and collagen deposition.

## 4. Materials and Methods

### 4.1. Construction of the Fusion Protein

hCol-ELP is a fusion protein that consists of the hCol and an elastin-like peptide (ELP). The Val-Pro-Gly-Xaa-Gly pentapeptide, in which Xaa denotes a 4:1 ratio of Val and Ala, was repeated in the ELP. To synthesize a vector that would encode the hCol-ELP fusion protein, the ELP was connected to the C-terminal of hCol through a (G_4_S)_3_ linker, such that the vector encoded hCol-ELP. Additionally, hCol and ELP were prepared as controls. The hCol, ELP, and hCol-ELP sequences were then ligated into the pET-28a vector. The designed sequences were optimized for codon preference, and GenScrip Biotechnology Co., Ltd. (Nanjing, China) synthesized these sequences and vectors.

### 4.2. Protein Expression and Purification

For expression, *E. coli* BL21(DE3) was transformed using plasmid constructs encoding hCol, ELP, and hCol-ELP. A single bacterial colony was selected for an overnight culture in 5 mL of kanamycin-containing Luria-Bertani (LB) medium. Until the culture density reached an OD600 of approximately 0.6, a 500 mL LB medium containing 100 μg/mL kanamycin was inoculated with the overnight bacterial culture. Then, IPTG was added to a final concentration of 0.5 mmol/L to enhance protein expression and the bacteria were cultured at 37 °C for 5 h. The cells were centrifuged at 4500 rpm for 10 min at 4 °C after culture. SDS-PAGE with densitometry scanning was used to measure the concentration of the target protein.

After centrifugation, cells were re-suspended in 40 mL of buffer (20 mM sodium phosphate, pH 7.4, 500 mM NaCl) and then disrupted using the Ultrasonic Cell Disruptor at 300 W for 5 min, three times. The cell disruption liquid was then enriched by centrifugation at 8000 r/min and 4 °C for 30 min. With the addition of 3 M (NH4) 2SO4 to a final concentration of 0.75 M, enriched liquid was mixed and incubated at 4 °C for 3 h and centrifuged at 12,000 rpm for 5 min. The proteins were initially purified via the salting-out extraction method [52]. The protein was then further purified by ion-exchange chromatography using the BabyBio Q ion-exchange column. The column was equilibrated in 20 mM Tris, pH 8.0, and washed with 20 mM Tris, 1 M NaCl, pH 8.0. The outflow was then collected at different time points based on the position of the peaks using an AKTA system (GE Healthcare, Chicago, IL, USA). Finally, the yield and purity of the target protein were verified through high-performance liquid chromatography (HPLC) and SDS-PAGE analysis.

### 4.3. Characterization of the Fusion Protein

#### 4.3.1. Purity and Concentration of the Protein

Purified hCol, ELP, and hCol-ELP proteins were dissolved in PBS. BCA was used to calculate the protein concentrations, whereas HPLC and SDS-PAGE were used to assess the purity of the proteins. A NanoChrom BioCore SEC-150 column, a mobile phase consisting of 50 mM PB, 100 mM NaCl, and 10% ACN, and a flow rate of 0.8 mL/min were used in these tests. The test were carried out using a Beckman HPLC system.

#### 4.3.2. Morphology and Structure of Protein

The appearance and interior structures of hCol, ELP, and hCol-ELP protein lyophilized powders were observed by camera and scanning electron microscopy, respectively. The hCol, ELP, and hCol-ELP proteins were dried using a vacuum freeze-drier (Thermo Fisher Heto, Waltham, MA, USA). Then, the specimens were fixed onto a titanium plate and coated with gold. The samples were observed via a scanning electron microscope (SEM, Hitachi, Tokyo, Japan) at an accelerating voltage of 10 kV.

#### 4.3.3. Collagen Properties of Protein

The collagen properties of hCol and hCol-ELP were investigated in vitro by collagenase digestion tests (Sigma-Aldrich, St. Louis, MO, USA). A total of 50 μM of hCol or hCol-ELP protein solution was mixed with collagenase (500:1, molar ratio) and incubated at 37 °C. An amount of 40 μL was taken from solutions at time points of 0.2, 0.5, 1, 4, and 6 h and stored at −80 °C. ELP was used as a control. The collagenase digestion degree of the sample was monitored using SDS-PAGE analysis.

#### 4.3.4. Stability of Protein

In vitro stability testing was performed in PBS and tissue fluid systems. The PBS (pH 7.5) system was placed at 37 °C for 96 h and measured after 0, 48, and 96 h. The tissue fluid system was placed at 37 °C for 15 days and measured after 0, 3, 7, 11, and 15 days. The degradation degree was monitored via SDS-PAGE analysis.

### 4.4. In Vitro Assays of Function

#### 4.4.1. Proliferation Assay

NIH-3T3 cells were purchased from Boster Biological Technology Co., Ltd. (Wuhan, China) and grown at 37 °C in a 5% CO_2_-containing environment in Dulbecco’s Modified Eagle’s Medium (DMEM, Gibco, New York, NY, USA) enriched with 10% fetal bovine serum (FBS, Gibco, New York, NY, USA) and 1% penicillin/streptomycin (Gibco, New York, NY, USA). A total of 3000 cells/well were seeded into 96-well plates and incubated for the required amount of time. Afterward, the culture medium was changed to 100 μL of new media containing the test protein or PBS. The CCK8 (Seven Biotechnology Co., Ltd., Beijing, China) solution was added to each well after 24 h of culture. After 2 h of reaction, a 450 nm wavelength was used to measure the OD value.

#### 4.4.2. In Vitro Wound Healing Assay

NIH-3T3 cells (2.5 × 105 cells/well) were seeded into 12-well plates and grown to full confluence as a monolayer. The monolayer was lightly scratched at the center of the well with a sterile 1 mL pipette tip and washed with PBS twice to remove the isolated cells. Then, fresh DMEM medium containing 0.1% FBS and treatments (5 μM) was added to each well and then cells grown for 24 h. The images were observed and analyzed from the same fields at 0 h and 24 h using a microscope. Then, the scratch area was analyzed using Image J v2.6 software. The wound closure rate was calculated using the following formula:Wound closure (%) = (A_0_ − A_24_)/A_0_ × 100,
where A_0_ was the wound area at 0 h and A_24_ was the wound area at 24 h.

#### 4.4.3. Collagen I Regeneration Assay

HSF cells were purchased from Boster Biological Technology Co., Ltd. (Wuhan, China) and cultured in Modified Eagle’s Medium (MEM, Gibco, New York, NY, USA). Cells were seeded into 24-well plates and incubated until the density reached 80%. Then, the culture medium was replaced with 500 µL fresh medium containing TGFβ1 (10 ng/mL), hCol (5 µM), and hCol-ELP (5 µM) for 48 h [43]. After incubation, the supernatant was collected from the cell culture and soluble pro-collagen I levels were quantified using a human pro-collagen I α1 ELISA kit (cat#: EHC083aQT) according to manufacturer’s instructions (Neobioscience, Shenzhen, China). Following the removal of the cells, the separated HSF was subjected to total RNA extraction in accordance with the manufacturer’s instructions (ES Science, Shanghai, China). After eliminating genomic DNA following the manufacturer’s protocol (Takara, Maebashi, Japan), the total RNA obtained underwent reverse transcription to generate cDNA. In a LightCycler 480 II device from Roche in Basel, Switzerland, a reverse transcription polymerase chain reaction (RT-PCR) was performed using 2 TB Green Premix Ex Taq II from Takara in Japan. The 40 PCR cycles were performed under the following thermal cycling conditions: initial denaturation at 95 °C for 10 min, followed by annealing at 60 °C for extension at 72 °C, denaturation at 95 °C for 15 s, and finally a final extension at 95 °C for 1 min. For the purpose of calculating the percentage of each target gene, GAPDH was used as an internal control. Table 3 displays the primer sequences.

### 4.5. In Vivo Wound Healing Assay

#### 4.5.1. Skin Wound Healing Model in Mice

The “3Rs” principle was adhered to in all animal care and usage programs employed in the study, which were carried out in accordance with the Regulations for the Administration of Affairs Concerning Experimental Animals set by the State Council of the People’s Republic of China. Forty Kunming (KM) male mice (7–8 weeks; 20–25 g) were purchased from SPF Biotechnology Co., Ltd. (Beijing, China), randomly divided into 4 groups, and housed individually in specific-pathogen-free conditions for one week. Prior to the procedure, the mice’s back hair was shaved, ethanol was swabbed, and anesthesia was administered. Full-thickness circular wounds with diameters of 10 mm were made on the back of the mice using a disposable biopsy punch. The wound was then covered with hCol, hCol-ELP, or ELP membranes (1 mg/membranes) and fixed on the wound with a 3M clear compress. As a negative control group, a group of mice that did not receive any treatment for their wounds was used. All of the mice cleaned their wounds on the third day and received the medication once again. The wound was monitored and recorded using a camera on days 0, 3, 5, 7, 11, and 14. Image Pro Plus 6.0 software was used to measure the area of the wound at each time point. The wound healing rate was calculated using the following formula:Wound closure (%) = (1 − A_n_/A_0_) × 100,
where A_0_ was the initial wound area and A_n_ was the area of wound on days 0, 3, 5, 7, 11, or 14 [53].

#### 4.5.2. Histological Analysis

On the 7th and 14th days after the operation, half of the animals were killed, the skin of the wound was collected, and the healing of the wound was evaluated at the histological level. Half of each skin tissue was then fixed in 10% formalin (VWR) and stained with Masson trichrome and H&E (Solarbio Science & Technology Co., Ltd., Beijing, China). Finally, the sections were observed and photographed under a Leica Application Suite Image 4.0 System. Image Pro Plus was used to determine the remaining wound width, epidermal thickness, and collagen deposition at the wound site.

#### 4.5.3. Immunofluorescence Staining Analysis

The sections were washed with PBS and then fixed with 4% formaldehyde for 30 min. Then, they were incubated with the collagen I antibody (Proteintech Group, Inc., Wuhan, China) working solution diluted with PBS at 4 °C overnight. On the glass slide, the working solution of the mixed fluorescent secondary antibody (Solarbio) was dropped and diluted with PBS. It was then incubated for 1 h at room temperature in the dark, washed again with PBS and the film sealed with DAPI (Beyotime Institute of Biotechnology, Beijing, China) staining solution dropwise. The sections were photographed using a fluorescence microscope (Leica, DM6000B, Wetzlar, Germany). Image Pro Plus was used to determine the relative fluorescence intensity and area.

#### 4.5.4. RT-PCR Analysis

The relative mRNA expression levels of each gene in the wound tissue were determined using RT-PCR analysis. According to the manufacturer’s instructions (ES Science, Shanghai, China), total RNAs were isolated from the skin of the wound. The resulting total RNA was reverse transcribed to produce cDNA after removing genomic DNA according to the manufacturer’s protocol (TakaRa, Japan). q-PCR was performed in a fluorescent quantitative PCR instrument (ROCGENE, Archimed-X4, Beijig, China). The primer sequences used are presented in Table 4.

### 4.6. Statistical Analysis

GraphPad Prism 8.0 (GraphPad Software Inc., La Jolla, CA, USA) was used for statistical analysis. Data were analyzed using an unpaired *t*-test or one-way ANOVA and are presented as the mean ± standard deviation (SD). Results were considered significant at *p* ≤ 0.05.

## 5. Conclusions

In this study, we successfully designed and synthesized the fusion molecule hCol-ELP, which contains ELP and hCol. We found that the stability of the hCol-ELP fusion protein was significantly better than that of free hCol. In addition, hCol-ELP can accelerate wound healing in mouse skin wounds by promoting the rapid proliferation of epithelial cells, decrease expression of TNF-α and IL-6 to reduce inflammation, improve collagen deposition and reorganization, and shorten scab formation time, especially in the early stages of trauma (hemostasis and inflammation). Therefore, the hCol-ELP fusion protein may have broad clinical applications in skin wound repair and scar inhibition. Additionally, the exact molecular signaling mechanism of action of hCol-ELP in wound healing may be further explored.

## Figures and Tables

**Figure 1 molecules-28-06773-f001:**
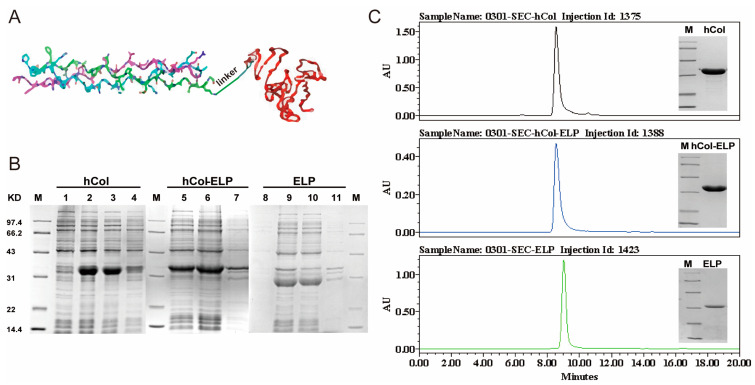
Successful construction and expression of the fusion protein hCol-ELP. (**A**) Structures of the hCol-ELP fusion proteins. (**B**) Expression of hCol, hCol-ELP, and ELP analyzed using SDS-PAGE. Lanes 1 and 8: bacteria pre-induction. Lanes 2, 5, and 9: bacteria following IPTG induction. Lanes 3, 6, and 10: supernatant after cell disruption. Lanes 4, 7, and 11: pellet after crushing. (**C**) Purified hCol, hCol-ELP, and ELP proteins were analyzed using SDS-PAGE and size exclusion chromatography. The purities of the hCol, hCol-ElP, and ELP proteins were 93.1%, 97.5% and 99.1%, respectively.

**Figure 2 molecules-28-06773-f002:**
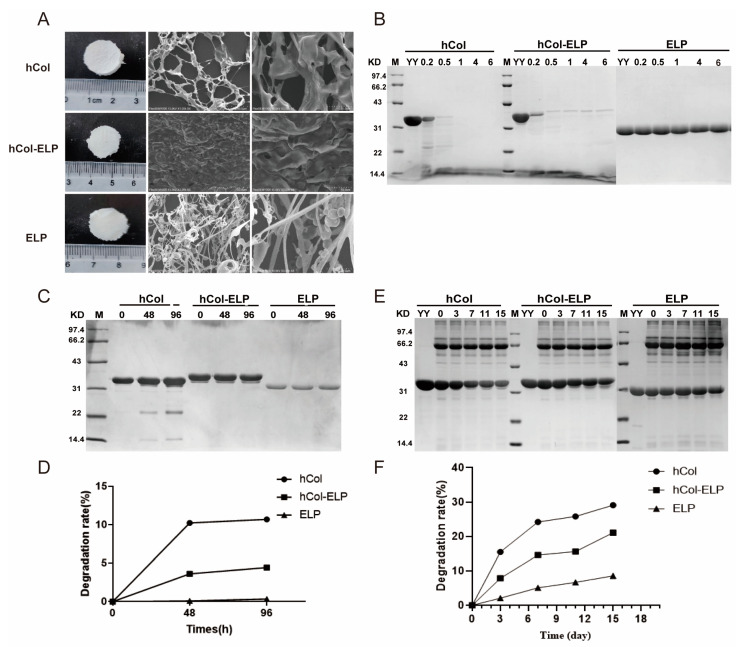
Characterizations of hCol-ELP. (**A**) Macroscopic view (**Left**) and SEM images (**Right**) of hCol, hCol-ELP, and ELP. (**B**) Enzymatic degradation profiles of hCol, hCol-ELP, and ELP. (**C**,**D**) Accelerated stability identification and degradation rates of the hCol, hCol-ELP, and ELP proteins in PBS systems. (**E**,**F**) Accelerated stability identification and degradation rates of the hCol, hCol-ELP, and ELP proteins in tissue fluid systems. M: molecular weight (MW) ladder, YY: protein stock solution without tissue fluid.

**Figure 3 molecules-28-06773-f003:**
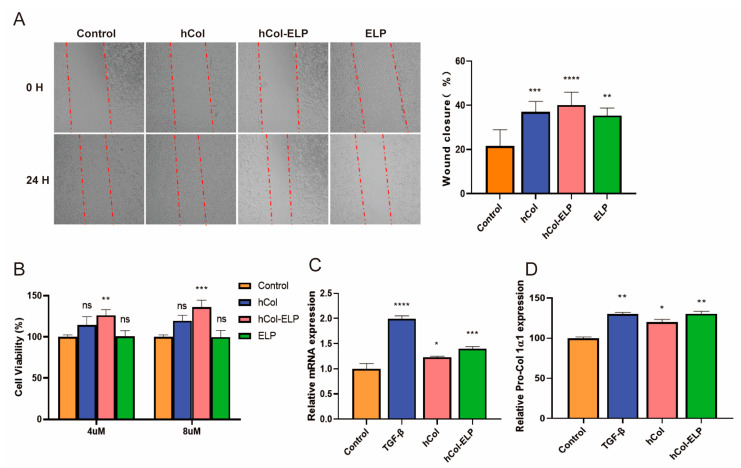
The proliferation, migration, and regeneration of collagen I in vitro. (**A**) Representative images of cell migration and wound closure rate in NIH/3T3 cells after culturing with hCol, hCol-ELP, and ELP for 24 h. (**B**) Proliferation of NIH/3T3 cells in different groups. (**C**) The expression of pro-collagen I α1 was detected by ELISA after 48 h of treatment with transforming growth factor beta (TGF-β), hCol, and hCol-ELP. (**D**) The relative mRNA expression of collagen I α1 was detected by RT-PCR after 48 h of treatment with TGF-β, hCol, and hCol-ELP. (*n* = 3, * *p* < 0.05, ** *p* < 0.01, *** *p* < 0.001, **** *p* < 0.0001, ns > 0.05 vs. control group).

**Figure 4 molecules-28-06773-f004:**
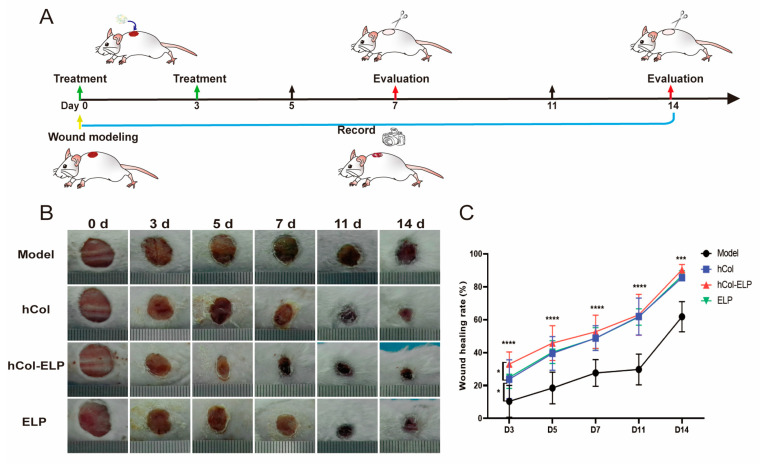
Skin wound healing over time in mice treated with hCol-ELP. (**A**) Schematic illustration of the experimental procedure. (**B**) Representative wound images from each group at 0, 3, 5, 7, 11, and 14 days. (**C**) Wound healing rate of each group at different times (on days 0, 3, and 5, *n* = 10; on days 7, 11, and 14, *n* = 5; * *p* < 0.05, *** *p* < 0.001, **** *p* < 0.0001 vs. model. All data are presented as mean ± SD).

**Figure 5 molecules-28-06773-f005:**
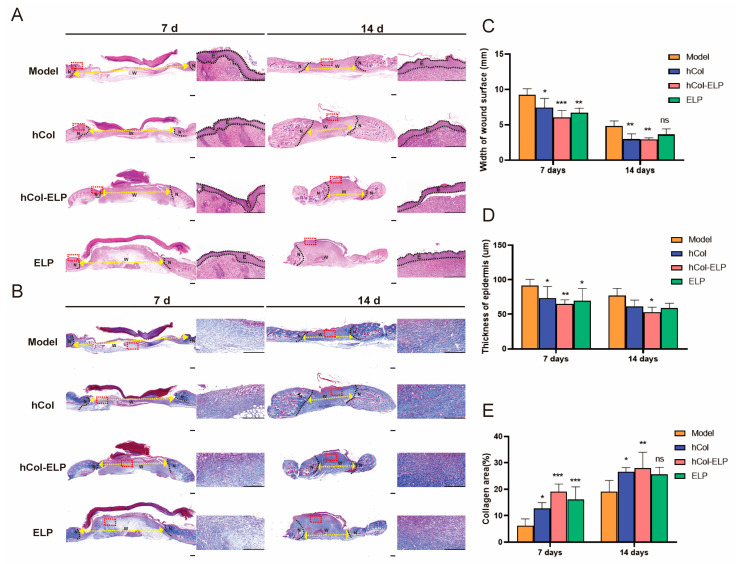
Histological and cytokine analysis of wound areas on the 7th and 14th days. (**A**,**B**) Hematoxylin–eosin staining (H&E) and Masson staining of tissue sections. The dotted line is the interface between the wound area and neo-epidermis. The yellow arrows indicate remaining wound length. W, wound surface; N, neo-epidermis; E, epidermis. Scale bars = 500 and 20 μm; magnification: 20× and 400×. (**C**) Width of the wound surface. (**D**) Thickness of regenerated epidermis. (**E**) The statistical analysis of collagen I in the wound area. (*n* = 5, * *p* < 0.05, ** *p* < 0.01, *** *p* < 0.001, vs. model. All data are presented as mean ± SD).

**Figure 6 molecules-28-06773-f006:**
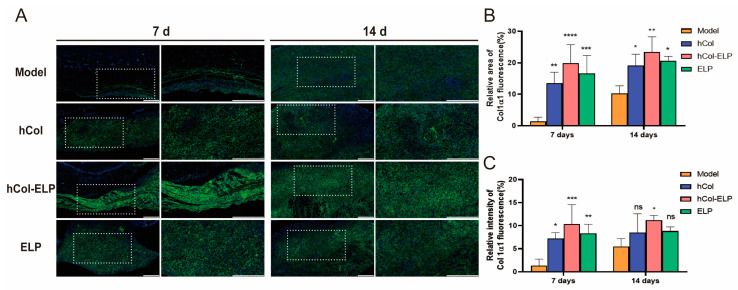
The collagen I staining of the wounded skin. (**A**) The collagen I staining of each group was assessed after 7 and 14 days of wound healing. The white dotted line represents the enlarged area. Scale: 50 and 20 μm; magnification: 200 and 400×. (**B**,**C**) The statistical analysis of collagen I in the wound area on the 7th and 14th days of treatment. (*n* = 5, * *p* < 0.05, ** *p* < 0.01, *** *p* < 0.001, **** *p* < 0.0001 vs. model. All data are presented as mean ± SD).

**Figure 7 molecules-28-06773-f007:**
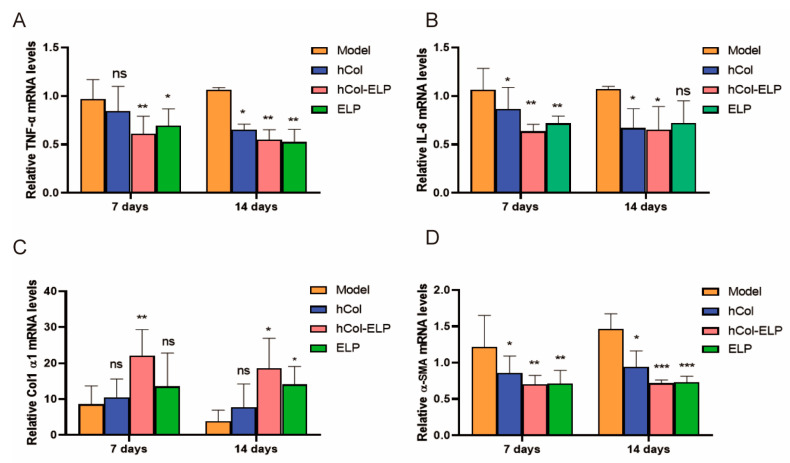
Relative mRNA expressions of TNF-α, IL-6, Col1 α1, and α-SMA on the 7th and 14th days. (**A**) Relative TNF-α mRNA levels. (**B**) Relative IL-6 mRNA levels. (**C**) Relative Col1 α1 mRNA levels. (**D**) Relative α-SMA mRNA levels. (*n* = 5, * *p* < 0.05, ** *p* < 0.01, *** *p* < 0.001, vs. model. All data are presented as mean ± SD).

**Figure 8 molecules-28-06773-f008:**
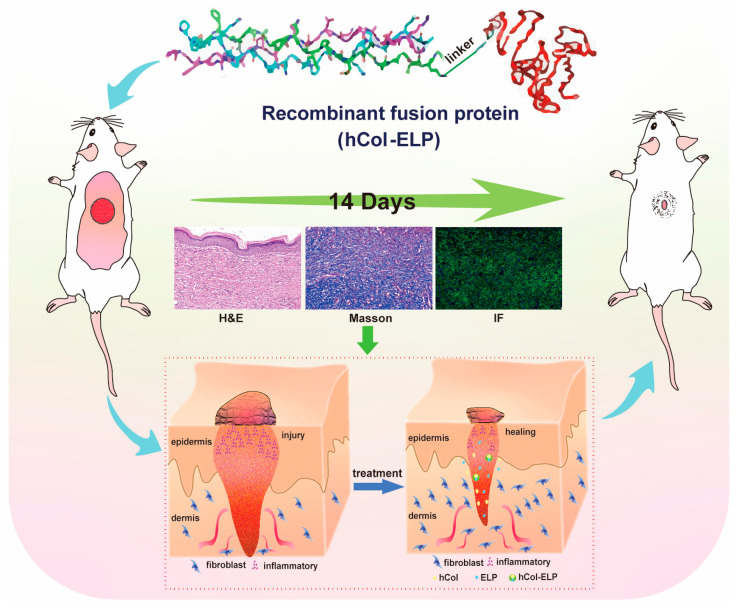
Construction of the hCol-ELP fusion protein and evaluation of its performance in promoting wound healing. hCol-ELP can accelerate wound healing and prevent scar hyperplasia by reducing inflammation and promoting collagen regeneration in mouse models of skin wounds.

**Table 1 molecules-28-06773-t001:** Wound closure rate of mice.

Group	Wound Closure Rate				
	Day 3	Day 5	Day 7	Day 11	Day 14
Model	10.39 ± 9.70%	18.48 ± 9.67%	27.62 ± 8.11%	29.75 ± 9.35%	61.81 ± 9.17%
hCol	23.82 ± 11.86% *	39.53 ± 10.26% *	48.79 ± 7.58% *	61.88 ± 11.28% *	85.47 ± 1.96% *
ELP	24.96 ± 6.77% *	40.25 ± 6.93% *	48.70 ± 6.43% *	61.60 ± 4.95% *	86.72 ± 2.99% *
hCol-ELP	33.02 ± 7.34% ****	45.77 ± 10.61% ****	52.64 ± 10.14% ****	62.95 ± 12.45% ****	90.36 ± 3.24% ***

* *p* < 0.05, *** *p* < 0.001, **** *p* < 0.0001 vs. model. All data are presented as mean ± SD.

**Table 2 molecules-28-06773-t002:** The width of the wound surface and thickness of epidermis.

Group	The Width of Wound Surface		The Thickness of Epidermis	
	Day 7	Day 14	Day 7	Day 14
Model	9.26 ± 0.88 mm	4.84 ± 0.72 mm	91.51 ± 9.03 μm	77.13 ± 10.49 μm
hCol	7.46 ± 1.29 mm *	2.98 ± 0.76 mm **	73.40 ± 16.96 μm *	61.65 ± 8.92 μm
ELP	6.72 ± 0.64 mm **	3.63 ± 0.82 mm	69.80 ± 17.74 μm *	59.16 ± 6.85 μm
hCol-ELP	6.08 ± 0.97 mm ***	2.94 ± 0.22 mm **	65.15 ± 5.75 μm **	53.33 ± 6.99 μm *

* *p* < 0.05, ** *p* < 0.01, *** *p* < 0.001 vs. model. All data are presented as mean ± SD.

**Table 3 molecules-28-06773-t003:** Primers used in real-time PCR experiments in vitro (designed by Jinsilui Biotechnology Co., Ltd.,Nanjing, China).

Gene Name	Primer Forward (5′-3′)	Primer Reverse (3′-5′)
COL1A1	GCCAAATATGTGTCTGTGACTCA	GGGCGAGTAGGAGCAGTTG
GAPDH	CCTGCCTCTACTGGCGCTGC	GCAGTGGGGACACGGAAGGC

**Table 4 molecules-28-06773-t004:** Primers used in real-time PCR experiments in vivo (designed by Jinsilui Biotechnology Co., Ltd., Nanjing, China).

Gene Name	Primer Forward (5′-3′)	Primer Reverse (3′-5′)
GAPDH	GCCCAGAACATCATCCCTGCAT	GCCTGCTTCACCACCTTCTTGA
TNF-α	GGTGCCTATGTCTCAGCCTCTTC	TGATCTGAGTGTGAGGGTCTGGG
IL-6	GGATACCACTCCCAACAGACCTG	TGTTCTTCATGTACTCCAGGTAGCT
COL1α1	AGAGCGGAGAGTACTGGATCGAC	GGGAATCCATCGGTCATGCTCTC
α-SMA	GATGCAGAAGGAGATCACAGCCC	CCCAGCTTCGTCGTATTCCTGTT

## Data Availability

All data are available in the main text.

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
