# Peer review of "Construction of a Collagen-like Protein Based on Elastin-like Polypeptide Fusion and Evaluation of Its Performance in Promoting Wound Healing"

_molecules, 2023, doi:10.3390/molecules28196773_

Round 1
Reviewer 1 Report
Chen et al. explain their newly designed hCol-ELP improves the stages of wound healing and inflammation in the wound sites. This study has merits.
- Authors may include a paragraph in the introduction stating the normal wound healing process and factors affecting wound healing (Ref: PMID: 20139336, PMID: 22028946 and PMID: 30627050), then mention collagen and elastin and their significance in wound healing (PMID: 22279382, PMID: 32602815, PMID: 23109320, etc.)
- Discussing wound healing, collagen, etc., may be unfit (lines 330, 264, 253, etc.). Discuss directly with your results instead of giving an introduction in every first sentence of the discussion.
- The significance of scar formation in terms of hCol-ELP compared to other established products may be discussed.
- Line 16: ‘called’ may be replaced with ‘named’ as it is a new abbreviation.
- Some reference in the introduction and discussion states the same information. Avoid repetition.
- A conclusion paragraph on what stages hCol-ELP significantly promotes wound healing was explained. The TNF-α and IL-6 expressions may be mentioned, stating their role in improving inflammation at the wound site. Also, note the exact molecular signaling mechanism of action of hCol-ELP in wound healing may be further explored.
- Figure 4: A recent publication explored the same pattern of wound healing over time, and the wound closure rate was calculated (Hemtong et al.; Future Pharmacology, 2023). This may be referenced in the material and methods. The authors may also explain the similarities/differences in the time points adopted in both studies (one in rats and the other in mice).
- Line 473: 40 may be represented as ‘Forty’
- The primer design of IL-6, Col1 1a, a-SMA, and GAPDH shows correct; however, the TNF-α was doubtful in the primer blast. The authors may please recheck the primer design.
- Abbreviations for the TNF, IL-6, Col1, SMA, etc., may be mentioned in the first instance in the manuscript.
- The average weight of animals may be mentioned with the age
- The ethical committee approval number and details may be included in the manuscript.
Author Response
Dear Professor Wang and reviewers:
Thanks for your comments and the extension in time for editing our manuscript “Construction of a Collagen-like Protein Based on Elastin-like-Polypeptide Fusion and Evaluation of Its Performance in Promoting Wound Healing” (molecules-2585666). Those comments are all valuable and very helpful for revising and polishing our paper, as well as the important guiding significance to our researches. We have studied comments carefully and have made correction which we hope meet with approval.
The main corrections in the paper and the responds to the reviewer’s comments are as flowing:
Reviewer #1
- Authors may include a paragraph in the introduction stating the normal wound healing process and factors affecting wound healing (Ref: PMID: 20139336, PMID: 22028946 and PMID: 30627050), then mention collagen and elastin and their significance in wound healing (PMID:22279382, PMID: 32602815, PMID: 23109320, etc.)
Reply: Thanks for your suggestion. According to your opinion, we collected all of this literature and integrated it into the introduction, supplementing the content in the introduction section. The skin, which is made up of the epidermis, dermis, and subcutaneous tissue, is the body's largest organ system [1]. The integrity of healthy skin has three main functions: barrier function, sensory function, and metabolic function [2]. Physiological regulation of skin wound healing is a complex and overlapping process, it is mainly composed of four phases: the hemostasis, the inflammatory, the proliferative, the remodeling phase [3]. However, there are many factors that affect wound healing, such as: age, medication, obesity, alcohol consumption, smoking, nutrition, etc [4], the most advanced treatment for wound healing is gels rich in the patient's platelet plasma (PRP) [5,6], but the treatment is expensive and has some side effects. According to current reports, collagen has the poten-tial to accelerate wound healing, elastin has the effect of preventing scar proliferation [7,8], Skin transplantation rich in collagen and elastin can treat extensive skin wounds [9].
- Takeo, M.; Lee, W.; Ito, M. Wound healing and skin regeneration. Cold Spring Harb Perspect Med 2015, 5, a023267, doi:10.1101/cshperspect.a023267.
- Sorg, H.; Tilkorn, D.J.; Hager, S.; Hauser, J.; Mirastschijski, U. Skin Wound Healing: An Update on the Current Knowledge and Concepts. Eur Surg Res 2017, 58, 81-94, doi:10.1159/000454919.
- Li, J.; Zhou, C.; Luo, C.; Qian, B.; Liu, S.; Zeng, Y.; Hou, J.; Deng, B.; Sun, Y.; Yang, J.; et al. N-acetyl cysteine-loaded graphene oxide-collagen hybrid membrane for scarless wound healing. Theranostics 2019, 9, 5839-5853, doi:10.7150/thno.34480.
- Boucek, R.J. Factors affecting wound healing. Otolaryngol Clin North Am 1984, 17, 243-264.
- Carter, M.J.; Fylling, C.P.; Parnell, L.K. Use of platelet rich plasma gel on wound healing: a systematic review and meta-analysis. Eplasty 2011, 11, e38.
- Chuncharunee, A.; Waikakul, S.; Wongkajornsilp, A.; Chongkolwatana, V.; Chuncharunee, L.; Sirimontaporn, A.; Rungruang, T.; Sreekanth, G.P. Invalid freeze-dried platelet gel promotes wound healing. Saudi Pharm J 2019, 27, 33-40, doi:10.1016/j.jsps.2018.07.016.
- Sarangthem, V.; Singh, T.D.; Dinda, A.K. Emerging Role of Elastin-Like Polypeptides in Regenerative Medicine. Adv Wound Care (New Rochelle) 2021, 10, 257-269, doi:10.1089/wound.2019.1085.
- Almine, J.F.; Wise, S.G.; Weiss, A.S. Elastin signaling in wound repair. Birth Defects Res C Embryo Today 2012, 96, 248-257, doi:10.1002/bdrc.21016.
- Wollina, U. One-stage Reconstruction of Soft Tissue Defects with the Sandwich Technique: Collagen-elastin Dermal Template and Skin Grafts. J Cutan Aesthet Surg 2011, 4, 176-182, doi:10.4103/0974-2077.91248.
- Discussing wound healing, collagen, etc., may be unfit (lines 330, 264, 253, etc.). Discuss directly with your results instead of giving an introduction in every first sentence of the discussion.
Reply: Thanks for your comment. According to your opinion, we have deleted these parts.
- The significance of scar formation in terms of hCol-ELP compared to other established products may be discussed.
Reply: Thanks for your suggestion. According to your opinion, we have added a section to the discussion: Recombinant human collagen type III has been widely used in the field of medicine and aesthetics, and has a good therapeutic effect on promoting wound healing [49], but there is no specific report on inhibiting scar. Currently, there are products that use elastin peptides to delay skin aging [50]. In addition, elastin plays an important role in preventing scar hyperplasia [7,8], we combined the functions of collagen and elastin to create hCol-ELP fusion protein, the results show that hCol-ELP fusion molecule has a good effect on pro-moting wound healing and inhibiting scar.
- Dong, Z.; Liu, Q.; Han, X.; Zhang, X.; Wang, X.; Hu, C.; Li, X.; Liang, J.; Chen, Y.; Fan, Y. Electrospun nanofibrous membranes of recombinant human collagen type III promote cutaneous wound healing. J Mater Chem B 2023, 11, 6346-6360, doi:10.1039/d3tb00438d.
- Shiratsuchi, E.; Nakaba, M.; Yamada, M. Elastin hydrolysate derived from fish enhances proliferation of human skin fibroblasts and elastin synthesis in human skin fibroblasts and improves the skin conditions. J Sci Food Agric 2016, 96, 1672-1677, doi:10.1002/jsfa.7270.
- Line 16: ‘called’ may be replaced with ‘named’ as it is a new abbreviation.
Reply: Thanks for your suggestion. According to your opinion, we have modified the statement: Based on the above, we have developed a recombinant fusion protein named hCol-ELP, which consists of hCol and an elastin-like peptide (ELP).
- Some reference in the introduction and discussion states the same information. Avoid repetition.
Reply: Thanks for your suggestion. According to your opinion, duplicate parts of discussion have been deleted.
- A conclusion paragraph on what stages hCol-ELP significantly promotes wound healing was explained. The TNF-α and IL-6 expressions may be mentioned, stating their role in improving inflammation at the wound site. Also, note the exact molecular signaling mechanism of action of hCol-ELP in wound healing may be further explored.
Reply: Thanks for your suggestion. According to your opinion, we have modified the conclusion. See below for details: In this study, we successfully designed and synthesized the fusion molecule hCol-ELP, which contains ELP and hCol. We found that the stability of the hCol-ELP fusion protein was significantly better than that of free hCol. In addition, the hCol-ELP can accelerate wound healing in mouse skin wounds by promoting rapid proliferation of epithelial cells, Decreased expression of TNF-α and IL-6 to reduce inflammation, improving collagen deposition and reorganization, and quickly shortening scab formation time, especially in the early stages of trauma(hemostasis and inflammation). Therefore, the hCol-ELP may have broad clinical applications in skin wound repair and scar inhibition. Also, the exact molecular signaling mechanism of action of hCol-ELP in wound healing may be further explored.
- Figure 4: A recent publication explored the same pattern of wound healing over time, and the wound closure rate was calculated (Hemtong et al.; Future Pharmacology, 2023). This may be referenced in the material and methods. The authors may also explain the similarities/differences in the time points adopted in both studies (one in rats and the other in mice).
Reply: Thanks for your suggestion. According to your opinion, the publication has be referenced in the material and methods, we compare the two articles, In the article by Hemtong, the wounds of the rats were 6mm and almost healed in 7-12 days. In our study, the wounds of the mice were 10mm and took 14 days to heal. It is important to note that the size of the wound and the subjects used in the study were different. According to the report by Afshar et al., the healing rate of 2cm wounds in rats on the 14th day was similar to that of our mice (PMID: 37504912). Similarly, Wakabayashi et al. conducted a 10 mm wound healing experiment on mice and observed healing on the 14th day (PMID: 31985636). That indicates that rats have a better self-healing ability than mice.
- Line 473: 40 may be represented as ‘Forty’
Reply: Thanks for your suggestion. According to your opinion, we have modified the statement.
- Some reference in the introduction and discussion states the same information. Avoid repetition.
Reply: Thanks for your suggestion. According to your opinion, we examined the raw data and found that two TNF-α primers were used before the experiment, the results ware similar, and due to personal negligence, the corresponding TNF-α primers were this one: mus-TNF-α F: GGTGCCTATGTCTCAGCCTCTTC, R: TGATCTGAGTGTGAGGGTCTGGG.
- Abbreviations for the TNF, IL-6, Col1, SMA, etc., may be mentioned in the first instance in the manuscript.
Reply: Thanks for your suggestion. According to your opinion, we have modified the results section.
- The average weight of animals may be mentioned with the age
Reply: Thanks for your suggestion. According to your opinion, we have revised the relevant content, as detailed below: Forty Kunming (KM) male mice (7-8 weeks), 20-25g were purchase from SPF Biotechnology Co., Ltd. (Beijing, China).
- The ethical committee approval number and details may be included in the manuscript.
Reply: Thanks for your suggestion. According to your opinion, we have modified the institutional review board statement: All animal procedures were approved by the Institutional Animal Care Research Committee of Contec Medical Systema Co., Ltd (M0.2023-04-10-05 IACUC Issue No.).
We tried our best to improve the manuscript and made some changes in the manuscript. These changes will not influence the content and framework of the paper. We appreciate for your warm work earnestly, and hope that the correction will meet with approval.
If you have any further questions, please let me know and offer us an additional opportunity to revise the manuscript. Once again, thank you very much for your comments and suggestions.
Reviewer 2 Report
Comments in the file attached.

Author Response
Dear Professor Wang and reviewers:
Thanks for your comments and the extension in time for editing our manuscript “Construction of a Collagen-like Protein Based on Elastin-like-Polypeptide Fusion and Evaluation of Its Performance in Promoting Wound Healing” (molecules-2585666). Those comments are all valuable and very helpful for revising and polishing our paper, as well as the important guiding significance to our researches. We have studied comments carefully and have made correction which we hope meet with approval.
The main corrections in the paper and the responds to the reviewer’s comments are as flowing:
Reviewer #2
- Line 59-61: It is true that inadequate viral inactivation and sterilization processes can lead to the transmission of these pathogens but the sentence needs to be better contextualized by specifying that this is not a risk to be generalized given the narrow safety pathways that are implemented in manufacturing. In addition, it would be worth evaluating and possibly citing any case histories described in the literature. Indeed, such a sentence, if read out of its proper context and without specific references can be mistakenly interpreted.
Reply: Thanks for your suggestion. We agree with you wholeheartedly and have thoroughly consulted relevant literature to revise introduction statement. Moreover, collagen can significantly reduce scar contracture and the risk of dysfunction. However, the main component of collagen products commonly used in clinical practice is animal-derived collagen [24,25], and the US Food and Drug Administration clearly states that given the narrow safety pathways that are implemented in manufacturing,animal-derived materials maybe pose a risk of transmitting viruses and spongiform en-cephalopathy [26]. For example, the previous outbreak of the mosquito-borne flavivirus Zika virus (ZIKV) has posed a huge threat to the safety of biological agents [27].
- De Angelis, B.; Gentile, P.; Tati, E.; Bottini, D.J.; Bocchini, I.; Orlandi, F.; Pepe, G.; Di Segni, C.; Cervelli, G.; Cervelli, V. One-Stage Reconstruction of Scalp after Full-Thickness Oncologic Defects Using a Dermal Regeneration Template (Integra). Biomed Res Int 2015, 2015, 698385, doi:10.1155/2015/698385.
- Qiu, X.; Wang, J.; Wang, G.; Wen, H. Vascularization of Lando((R)) dermal scaffold in an acute full-thickness skin-defect porcine model. J Plast Surg Hand Surg 2018, 52, 204-209, doi:10.1080/2000656X.2017.1421547.
- Medical devices containing materials derived from animal sources (except for in vitro diagnostic devices), guidance for FDA reviewers and industry; availability--FDA. Notice. Fed Regist 1998, 63, 60009-60010.
- Zmurko, J.; Vasey, D.B.; Donald, C.L.; Armstrong, A.A.; McKee, M.L.; Kohl, A.; Clayton, R.F. Mitigating the risk of Zika virus contamination of raw materials and cell lines in the manufacture of biologicals. J Gen Virol 2018, 99, 219-229, doi:10.1099/jgv.0.000995.
- Materials and Methods The section is well structured and well detailed. I suggest moving the materials and methods section before the results. In fact, some acronyms are used in other sections without bringing back their extended form previously, e.g., HE and CCK8.
Reply: Thanks for your suggestion. According to your opinion, we have modified the acronyms, but the molecules magazine format is the materials and methods section after the results.
- 3.Results The description of the results is clear and detailed, and the statistical data are clearly reported, but it might be useful and easier to read by including them in special tables at the conclusion of each subsection instead of inserting them in the captions of the figures presented.
Reply: Thanks for your suggestion. According to your opinion, we have included the statistical data in special tables at the conclusion of each subsection.
Table 1. Wound closure rate of mice.
Group |
Wound closure rate |
||||
Day 3 |
Day 5 |
Day 7 |
Day 11 |
Day 14 |
|
Model |
10.39 ± 9.70% |
18.48 ± 9.67% |
27.62 ± 8.11% |
29.75 ± 9.35% |
61.81 ± 9.17% |
hCol |
23.82 ± 11.86%* |
39.53 ± 10.26%* |
48.79 ± 7.58%* |
61.88 ± 11.28%* |
85.47 ± 1.96%* |
ELP |
24.96 ± 6.77%* |
40.25 ± 6.93%* |
48.70 ± 6.43%* |
61.60 ± 4.95%* |
86.72 ± 2.99%* |
hCol-ELP |
33.02 ± 7.34%**** |
45.77 ± 10.61%**** |
52.64 ± 10.14%**** |
62.95 ± 12.45%**** |
90.36 ± 3.24%*** |
Table 2. The width of wound surface and the thickness of epidermis.
Group |
The width of wound surface |
The thickness of epidermis |
||
Day 7 |
Day 14 |
Day 7 |
Day 14 |
|
Model |
9.26 ± 0.88 mm |
4.84 ± 0.72 mm |
91.51 ± 9.03 μm |
77.13 ± 10.49 μm |
hCol |
7.46 ± 1.29 mm* |
2.98 ± 0.76 mm** |
73.40 ± 16.96 μm* |
61.65 ± 8.92 μm |
ELP |
6.72 ± 0.64 mm** |
3.63 ± 0.82mm |
69.80 ± 17.74 μm* |
59.16 ± 6.85 μm |
hCol-ELP |
6.08 ± 0.97 mm*** |
2.94 ± 0.22 mm** |
65.15 ± 5.75 μm** |
53.33 ± 6.99 μm* |
- 4.Discussion The discussion is well-developed, supported by adequate bibliography, and the logical connection is straightforward and understandable. A form clarification is made below. Line 343: The verb "to be" should be conjugated in the third person singular ("was").
Reply: Thanks for your suggestion. According to your opinion, we have modified the discussion statement.
- Conclusions The conclusions are clear and somewhat concise.
Reply: Thanks for your suggestion. According to your opinion, we have modified the conclusion:In this study, we successfully designed and synthesized the fusion molecule hCol-ELP, which contains ELP and hCol. We found that the stability of the hCol-ELP fusion protein was significantly better than that of free hCol. In addition, the hCol-ELP can accelerate wound healing in mouse skin wounds by promoting rapid proliferation of epithelial cells, Decreased expression of TNF-α and IL-6 to reduce inflammation, improving collagen deposition and reorganization, and quickly shortening scab formation time, especially in the early stages of trauma(hemostasis and inflammation). Therefore, the hCol-ELP may have broad clinical applications in skin wound repair and scar inhibition. Also, the exact molecular signaling mechanism of action of hCol-ELP in wound healing may be further explored.
We tried our best to improve the manuscript and made some changes in the manuscript. These changes will not influence the content and framework of the paper. We appreciate for your warm work earnestly, and hope that the correction will meet with approval.
If you have any further questions, please let me know and offer us an additional opportunity to revise the manuscript. Once again, thank you very much for your comments and suggestions.
Round 2
Reviewer 1 Report
I agree with the revised version of the manuscript.